# Osteosarcoma in Adolescents and Young Adults

**DOI:** 10.3390/cells10102684

**Published:** 2021-10-07

**Authors:** Jun Ah Lee, Jiwon Lim, Hye Young Jin, Meerim Park, Hyeon Jin Park, Jong Woong Park, June Hyuk Kim, Hyun Guy Kang, Young-Joo Won

**Affiliations:** 1Center for Pediatric Cancer, Department of Pediatrics, Goyang 10408, Korea; junahlee@ncc.re.kr (J.A.L.); hyjin@ncc.re.kr (H.Y.J.); meerim@ncc.re.kr (M.P.); hjpark@ncc.re.kr (H.J.P.); 2Division of Cancer Registration and Surveillance, Goyang 10408, Korea; mani98@ncc.re.kr; 3Orthopedic Oncology Clinic, Center for Rare Cancer, National Cancer Center, Goyang 10408, Korea; jwpark82@ncc.re.kr (J.W.P.); docjune@ncc.re.kr (J.H.K.); ostumor@ncc.re.kr (H.G.K.)

**Keywords:** osteosarcoma, adolescents and young adults (AYA), Korea

## Abstract

The epidemiology of osteosarcoma in adolescents and young adults (AYA) remains unclear. We aimed to assess and compare the clinical features of osteosarcoma between AYA and other age groups. We retrieved osteosarcoma cases diagnosed between 1999 and 2017 from the Korea Central Cancer Registry. We compared survival trends and clinical characteristics between AYA and other age groups. AYA comprised 43.3% (1309/3022) of the osteosarcoma cases. Compared to other age groups, the male-to-female ratio was highest in AYA (1.61:1). The proportion of tumors located in an extremity was 80.3% in AYA, which was lower than in young children (92.5%) or pubertal children (93.8%) but higher than in adults (55.7%) or the elderly (47.5%). As for treatments, 71.2% of AYA received local treatment and systemic chemotherapy, and 28.8% received only local treatment (surgery: 261, radiotherapy: 9, surgery and radiotherapy: 5). The 5-year overall survival (OS) was lower in AYA (68%) than in young children (78%) or pubertal children (73%) but higher than in adults (47%) or the elderly (25%). When AYA were divided into five subgroups by age, patients aged 15–19 years constituted the largest proportion (45.4%, *n* = 594). Additionally, the proportion of patients with a non-extremity tumor increased in an age-dependent manner, from 10.3% in AYA aged 15–19 years to 35.3% in AYA aged 35–39 years. OS did not significantly differ among the different age subgroups of AYA. The clinical characteristics and OS of the AYA were more similar to those of children than to those of adults. There is a need for cooperation between pediatric and adult oncologists for effective osteosarcoma treatment in AYA.

## 1. Introduction

Osteosarcoma is the most common primary malignant bone tumor in children and adolescents [1]. The prognostic significance of age in osteosarcoma remains unclear [2,3,4,5]. Studies have dichotomized patients using specific age-based cut-off values, followed by a comparison of survival rates [2,3,4,5]. Data from the Surveillance, Epidemiology, and End Results (SEER) database showed that patients aged greater than 15 years have a lower 5-year relative survival than those aged less than 15 years [6]. The Children Oncology Group (COG) reported that patients with osteosarcoma aged greater than 18 years have a significantly increased risk of relapse and death [7]. Given that the age range of 15 to 18 years corresponds to the beginning or middle of the adolescent period [8], there is a need to determine whether the outcomes of osteosarcoma in adolescents and young adults (AYA) are inferior to those of children.

The US Adolescent and Young Adult Oncology Progress Review Group defined AYA patients with cancer as those diagnosed with cancer between the ages of 15 and 39 years [9]. AYA are in a transitional period between different phases of life, with distinct epidemiological, clinical, and biological characteristics [9]. There remains scarce research on cancers in AYA [10]; moreover, current knowledge regarding osteosarcoma has been obtained from retrospective studies and clinical trials conducted by pediatric cooperative groups [11]. However, given the wide age span across AYA, these patients can receive clinical care from pediatric or adult oncologists. Although the clinical characteristics and outcomes of AYA with osteosarcoma remain unclear, population-based cancer registries could yield important insights.

Therefore, we aimed to analyze and compare the clinical features of osteosarcoma between AYA and other age groups using epidemiological data obtained from the Korea Central Cancer Registry (KCCR).

## 2. Materials and Methods

### 2.1. Data Sources

The KCCR contains information obtained from the entire Korean population with cancer since 1999, including demographics, date of first diagnosis, primary site, morphology, diagnostic method, stage, and initial treatment. We applied the osteosarcoma definition provided by the International Classification of Diseases for Oncology, 3rd edition [12]. We extracted data regarding the incidence and survival of patients with osteosarcoma between 1999 and 2017. Data regarding mortality were obtained from Statistics Korea [13]. This study was approved by the institutional review board of the National Cancer Center (IRB No. NCC 2021-0064).

### 2.2. Statistical Analyses

Patients were age-grouped as follows: young children, ≤9 years; pubertal children, 10–14 years; AYA, 15–39 years; adults, 40–64 years; and elderly individuals, ≥65 years. We performed among-group comparisons of the incidence and clinical characteristics, including sex, diagnosis period (1999–2003, 2004–2008, 2009–2013, or 2014–2017), tumor location (extremity: upper, lower; axial: pelvis, spine, or elsewhere), SEER summary stage (since 2006) [14], and the first treatment course within 4 months of diagnosis.

Overall survival (OS) rates were calculated using the Kaplan-Meier method. OS comparisons among patients grouped according to clinical variables were performed using the log-rank test. A Cox proportional hazards model was used to estimate the effects of covariates on hazard rates. All statistical tests were two-tailed. Statistical significance was set at *p* < 0.05. Statistical analyses were conducted using SAS software ver. 9.4 (SAS Institute Inc., Cary, NC, USA) and STATA software ver. 16 (StataCorp LLC, College Station, TX, USA).

## 3. Results

### 3.1. Osteosarcoma in the KCCR Database

Between 1999 and 2017, there were 3022 osteosarcoma cases in the national cancer registry database. The median age at osteosarcoma diagnosis was 21 years, with the teenage group showing the highest occurrence (Figure 1 and Table 1). Overall, there were more male patients (male-to-female ratio = 1.27:1) (Table 1). In most cases (2294 cases, 75.9%), the primary tumor site was an extremity.

Since 2006, the KCCR has included information regarding disease extent from 1972 cases. Among them, the disease extent at diagnosis was unknown and metastases in 413 (21.0%) and 648 (32.9%) cases, respectively. Treatment information was available for 2206 patients, with 2146 (97.3%) patients having undergone surgery. Multimodal treatment comprising both surgery and chemotherapy was administered to 1452 (65.8%) patients.

### 3.2. Comparisons of AYA Patients with Other Age Groups

Some typical clinical characteristics of osteosarcoma, including an extremity being the primary tumor site and male predominance, were evident in young children, pubertal children, and AYA. Contrastingly, in adults and elderly individuals, females comprised half of the cases; moreover, there was a higher proportion of patients with a non-extremity tumor location or metastasis at diagnosis (Table 2). There were differences in the administered treatment modalities among the age groups, with adults and elderly individuals having lower rates of systemic chemotherapy use. The proportions of patients who received only local treatment (surgery or radiotherapy) were 43.6% (205/470) and 73.5% (164/223) in adults and elderly individuals, respectively. The proportions of patients who received multimodal treatment (local treatment and systemic chemotherapy) were 56.4% (265/470) and 26.5% (59/223) among adults and elderly individuals, respectively. Contrastingly, 113 young children (83.7%), 368 pubertal children (87.2%), and 681 AYA (71.2%) received local treatment and systemic chemotherapy (Table 2).

AYA comprised 43.3% (1309) of the patients with osteosarcoma registered in the KCCR. The male-to-female ratio was higher in the AYA group (1.61:1) than in the other age groups. The proportion of extremity tumors was lower in AYA (80.3%) than in young children (92.5%) or pubertal children (93.8%), but higher than in adults (55.7%) or elderly individuals (47.5%). The proportion of patients with metastases at diagnosis in AYA was lower than that in adults or elderly individuals, but similar to that in younger patients. The proportion of patients who received multimodal treatment was lower in AYA (71.2%) than in young (83.7%) and pubertal children (87.2%). Moreover, the proportion of patients who only received local treatment was higher in AYA (28.8%) than in young children (16.3%) and pubertal children (12.8%) (Table 2).

The 5-year OS rates of AYA patients were similar between 1999 and 2013 (Figure 2).

Survival analysis of 3017 patients revealed a 5-year OS rate of 61%. OS was inversely correlated with age and abruptly decreased in adults (46%) and elderly individuals (25%) (Table 3). Univariate and multivariate analyses revealed that tumor location, presence of metastasis at diagnosis, and age were prognostic factors (Table 3).

Further, the OS rate in AYA was lower than that in young children or pubertal children, but higher than that in adults or elderly individuals. There was an approximately 5% difference in the OS rate among young children, pubertal children, and AYA (Table 3). Contrastingly, there was an approximately 20% difference in the OS rate among AYA, adults, and elderly individuals (68%, 47%, and 25%, respectively) (Table 3). AYA patients with extremity tumors showed a similar survival to young children and pubertal children (Figure 3A). The 5-year OS rates of patients with extremity tumors according to age were as follows: 77.9%, young children; 74.0%, pubertal children; 70.9%, AYA; 56.5%, adults; and 31.6%, the elderly. Additionally, the OS rate of AYA with axial tumors was lower than that of pubertal children (53.8% vs. 67.0%, Figure 3B).

### 3.3. Age-Based Subgroups Analysis of AYA Patients

Among five age-based subgroups of the AYA group, patients aged 15–19 years comprised the largest proportion (45.4%, *n* = 594). Moreover, there were among-subgroup differences in the clinical characteristics (Table 4). Patients aged 15–19 and 20–24 years showed male predominance, with male-to-female ratios of 1.9 (389 males, 205 females) and 2.3 (172 males and 76 females), respectively. Across all age subgroups, most tumors were located in the extremities. Further, there was an age-dependent increase in the proportion of non-extremity tumors as follows: 10.3%, 15–19 years; 16.5%, 20–24 years; 27.6%, 25–29 years; 36.8%, 30–34 years; and 35.3%, 35–39 years. There was no significant among-group difference in the OS rate (*p* = 0.08, Figure 4). However, patients aged 20–24 years showed the best 5-year OS rate (74.4%). Further, the OS rate in patients aged 15–19 years was slightly lower than that in patients aged 20–29 years.

## 4. Discussion

To our knowledge, this is the first study to report epidemiologic data regarding osteosarcoma in Korean AYA. AYA patients accounted for 43.3% of osteosarcoma cases registered in the KCCR from 1999 to 2017. The 5-year OS rate was 68% in Korean AYA patients with osteosarcoma, which fell in between the OS rates of children and adults. Among the AYA, there were slight age-based differences in the clinical characteristics and outcomes. For example, there was evident male predominance in patients aged <25 years; further, there was an increased proportion of non-extremity tumors in patients aged >30 years. There were no significant among-subgroup differences in the OS rates.

The clinical characteristics and survival rates of Korean AYA patients with osteosarcoma were more similar to those of children than those of adults. AYA have worse outcomes than children in various chemosensitive cancers [6]. This could be attributed to multiple factors, including delayed diagnoses, different tumor biology, decreased participation in clinical trials, worse compliance with chemotherapy schedules, distinct chemotherapy metabolism, and increased toxicity leading to delayed chemotherapy [15]. Several studies have reported that AYA patients with osteosarcoma have significantly worse survival rates than children [6,7]. For example, a COG study on osteosarcoma reported a significantly increased risk of relapse or death in AYA aged 18–30 years [7]. The EURAMOS study also reported that adolescents (males: 13–17 years; females: 12–16 years) and adults (males: ≥18 years; females: ≥17 years) had inferior event-free survival than had children (males: 0–12 years; females: 0–11 years) [16]. Further, the US SEER showed that the 5-year relative survival rate was lower in AYA (65.4%) than in children (<15 years; 75.6%) [6]. The survival patterns of Korean AYA patients with osteosarcoma were consistent with the US SEER data [6], falling between the survival rates of children and adults. Moreover, a study of the Japanese Bone and Soft Tissue Tumor registry reported similar survival rates in AYA (75.2%) and children (75.7%) [17]. However, Japanese AYA patients receive the same chemotherapy as children [17], suggesting that the survival rates for AYA could be improved to those in children by administration of the same meticulous multidisciplinary treatment.

Age-based subgroup analysis of the AYA patients revealed that the survival rate in patients aged 15–19-years was slightly lower than that in young children or pubertal children. Although there was no significant among-group difference in the survival rates, the 20–24-year-old subgroup showed the highest survival rate, which was similar to that in young and pubertal children. The reasons underlying these findings remain unclear. There were no differences in treatment modalities among AYA aged 15–19 years, children (<15 years), and AYA aged >20 years. The treatment compliance of AYA aged 15–19 years might be similar to that of children given the strong involvement of the parents or guardians in cancer treatment. Park et al. reported similar survival-related findings in Korean AYA patients with cancer; specifically, patients aged 15–19 years had lower survival rates than patients aged 20–39 years [18]. Therefore, there could be differences in tumor biology or drug metabolism between adolescents and young adults. There is a need for further studies to elucidate the reasons why patients aged 15–19 years have worse outcomes than those aged 20–24 years.

There were similar treatment modalities and survival rates of Korean AYA patients with osteosarcoma from 1999 to 2017. Unfortunately, specific treatment-related information, including the chemotherapy regimen or surgery extent, is unavailable in the KCCR database. However, the Korean Society of Pediatric Hematology and Oncology reported that >90% of patients with extremity tumors undergo limb salvage surgery [19]. Furthermore, systemic chemotherapy combining two to four agents is administered and is often postoperatively switched according to the histological response to preoperative chemotherapy [19]. Further, our findings indicated that surgery remained the standard for local treatment of osteosarcoma during the 19 year period. Additionally, the indications for radiotherapy are limited to osteosarcoma [19,20]. AYA patients who only received radiotherapy (2.1% [20/956]) had tumors located at sites where complete surgery was not feasible. Although information regarding the surgical margins was unavailable, 42 (4.4%) patients who underwent surgery and radiotherapy might have undergone incomplete surgery, with either a macroscopic or microscopic residual tumor. Excluding thyroid cancer, the 5-year relative survival rates of Korean AYA improved by 23.0% between 1993 and 1995 and 2012 and 2016 [18]. However, the survival rates of Korean patients with osteosarcoma, including AYA, were similar throughout the study period. This is consistent with previous clinical reports, indicating that the survival rates of patients with osteosarcoma have not changed over the past few decades [21].

This study has several limitations resulting from the nature of KCCR data. For example, the KCCR database lacks information regarding the previous medical history or comorbidities. Information on disease extent at diagnosis only became available after 2006; moreover, there remains no information regarding tumor size, tumor grade, chemotherapy regimen, and histopathological response to preoperative chemotherapy. Moreover, the registered treatment-related information only included the treatments administered during the first 4 post-diagnosis months. Therefore, the prognostic significance of these clinical variables should be considered only within the context of the currently available data.

In conclusion, the survival rate of Korean AYA patients with osteosarcoma was lower than that of children. Further, the clinical characteristics and outcomes slightly differed across age subgroups of AYA. Our findings indicate the need for collaboration between pediatric and adult oncologists to elucidate the biological characteristics and improve the outcomes of AYA with osteosarcoma.

## Figures and Tables

**Figure 1 cells-10-02684-f001:**
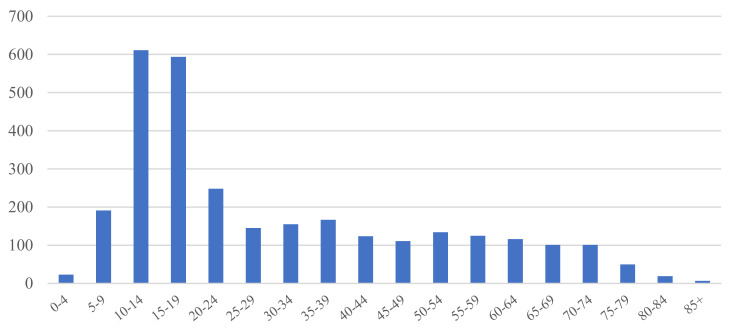
Age distribution of patients with osteosarcoma.

**Figure 2 cells-10-02684-f002:**
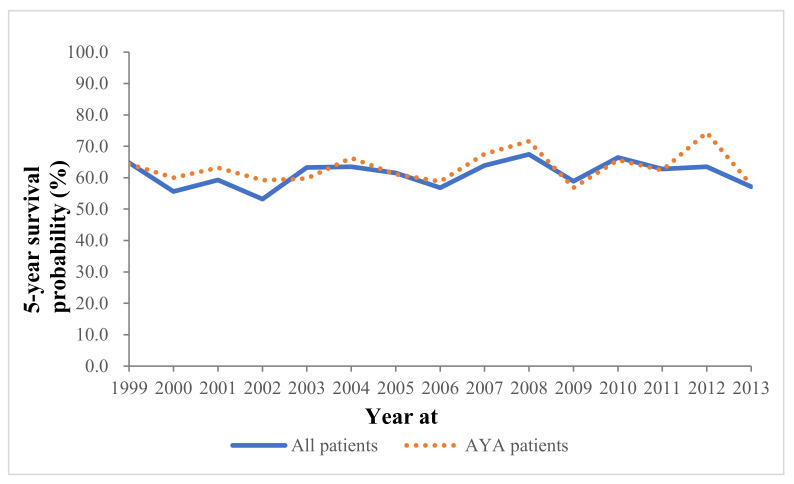
The 5-year overall survival trend of patients with osteosarcoma.

**Figure 3 cells-10-02684-f003:**
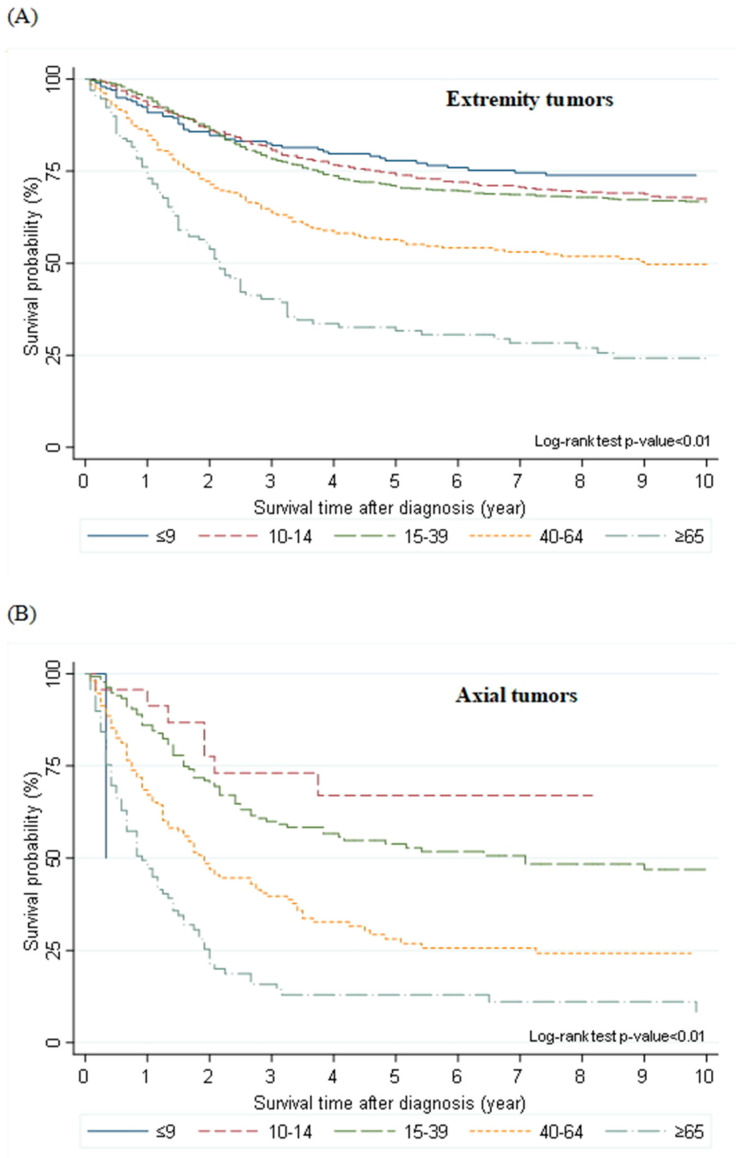
Overall survival of patients with osteosarcoma according to age and tumor site. (**A**) Extremity tumors, (**B**) Axial tumors.

**Figure 4 cells-10-02684-f004:**
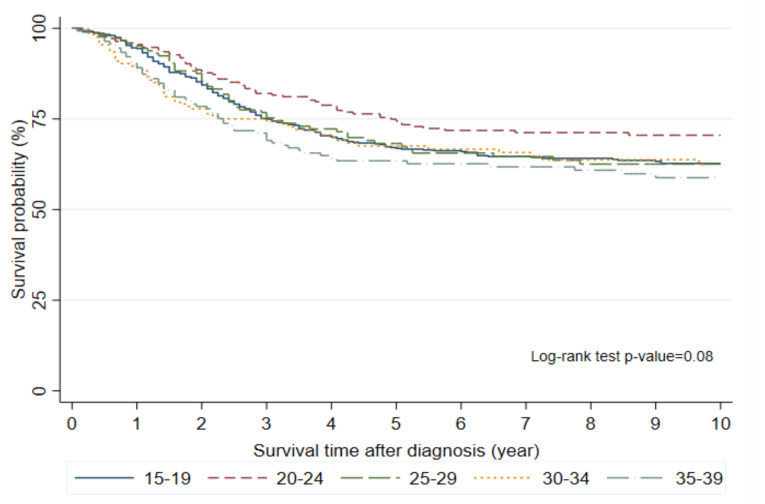
Overall survival of AYA by age subgroups.

**Table 1 cells-10-02684-t001:** Characteristics of osteosarcoma cases diagnosed between 1999 and 2017.

Characteristics		Number (*n* = 3022)	%
Sex	Male	1689	55.89
	female	1333	44.11
	Male to female ratio		1.27
Location	Extremity (upper, lower)	2294	75.91
	Axial (pelvis, spine)	404	13.37
	Elsewhere	324	10.72
Age group	Young children (≤9 years)	214	7.08
	Pubertal children (10–14 years)	611	20.22
	AYA (15–39 years)	1309	43.32
	Adults (40–64 years)	610	20.19
	Elderly (≥65 years)	278	9.20
Diagnostic period	1999–2003	764	25.28
	2004–2008	764	25.28
	2009–2013	812	26.87
	2014–2017	682	22.57
Extent of disease (*n* = 1972, since 2006)	Localized	911	46.20
	Metastasis	648	32.86
	Unknown	413	20.94
Treatment	Surgery only	650	21.51
	RT only	26	0.86
	Surgery + RT	44	1.46
	Surgery + CTx	1360	45.00
	CTx + RT	34	1.13
	Surgery + CTx + RT	92	3.04
	Unknown	816	27.00

**Table 2 cells-10-02684-t002:** Comparison of clinical characteristics between the age groups.

Variables		Age					
		Young Children (*n* = 214)	Pubertal Children (*n* = 611)	AYA (*n* = 1309)	Adults (*n* = 610)	Elderly (*n* = 278)	*p*-Value
Sex	Male	123	327	808	302	129	<0.0001
	Female	91	284	501	308	149	
	Male to female ratio	1.35	1.15	1.61	0.98	0.87	
Location	Extremity	198	573	1051	340	132	<0.0001
	Axial	2	23	138	151	90	
	Elsewhere	14	15	120	119	56	
Extent of disease (since 2006)	Localized	56	175	396	197	87	0.0010
	Metastasis	35	115	245	161	92	
	Unknown	33	103	159	82	36	
Treatment	Surgery only	22	51	261	185	131	<0.0001
	RT only	0	2	9	4	11	
	Surgery + RT	0	1	5	16	22	
	Surgery + CTx	109	356	633	214	48	
	CTx + RT	1	4	11	16	2	
	Surgery + CTx + RT	3	8	37	35	9	
	Unknown	79	189	353	140	55	

**Table 3 cells-10-02684-t003:** The overall survival and hazard ratio of osteosarcoma according to clinical variables.

Variables		No.	Median Survival Time (Month)	5-Year Survival (%) (CI)	Univariate		Multivariate	
					HR	*p*-Value	HR	*p*-Value
Sex	Male Female	1684 1333	- -	62 (59–64)	Reference	0.9198	Reference	0.1549
				62 (59–64)	1.01 (0.89–1.13)		0.92 (0.81–1.03)	
Location	Extremity	2291	-	68 (66–70)	Reference	<0.0001	Reference	<0.0001
	Axial	403	25	36 (31–41)	2.72 (2.33–3.17)		0.91 (1.62–2.24)	
	Elsewhere	323	54	49 (43–55)	1.77 (1.48–2.12)		1.31 (1.09–1.58)	
Extent of disease (since 2006)	Localized	910	-	73 (69–76)	Reference	<0.0001	Reference	<0.0001
	Metastasis	644	41	44 (40–48)	2.60 (2.19–3.10)		2.40 (2.03–2.84)	
	Unknown	413	-	71 (66–75)	0.98 (0.78–1.24)		1.57 (1.34–1.83)	
Age group	Young children	214	-	78 (71–83)	Reference	<0.0001	Reference	<0.0001
	Pubertal children	610	-	73 (70–77)	1.24 (0.90–1.70)		1.22 (0.89–1.67)	
	AYA	1307	-	68 (65–71)	1.47 (1.09–1.98)		1.35 (1.00–1.82)	
	Adults	608	44	47 (43–51)	3.06 (2.64–4.16)		2.55 (1.87–3.47)	
	Elderly	278	20	25 (20–31)	5.88 (4.28–8.08)		4.69 (3.39–6.49)	

**Table 4 cells-10-02684-t004:** Comparison of clinical characteristics among AYA age subgroups.

Variables		Age (Years)					
		15–19 (*n* = 594)	20–24 (*n* = 248)	25–29 (*n* = 145)	30–34 (*n* = 155)	35–39 (*n* = 167)	*p*-Value
Sex	Male	389	172	77	78	92	<0001
	Female	205	76	68	77	75	
Location	Extremity	533	207	105	98	108	<0001
	Axial	43	25	13	24	33	
	Elsewhere	18	16	27	33	26	
Extent of disease (since 2006)	Localized	171	71	49	51	54	0.7419
	Metastasis	110	45	30	27	33	
	Unknown	75	29	12	25	18	

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
