# Peer review of "Osteosarcoma in Adolescents and Young Adults"

_cells, 2021, doi:10.3390/cells10102684_

Round 1
Reviewer 1 Report
The strength of the current study is the population-based concept with no bias in patient selection and beyond the standard so-called classical osteosarcoma patients; non-metastatic, extremity localised tumor
The weakness is the limited data on treatment and the impact of less intensive chemotherapy on worse survival in non-children patients. Age is in general an adverse prognostic factor in oncology, as for osteosarcoma, the issue is whether poor outcome in older ostesarcoma patients is caused by improper chemotherapy or other treatment-related factors, that can be managed. Therefore, treatment related data including histological response to preoperative chemotherapy should be included to make the study relevant for the readers
Author Response
The strength of the current study is the population-based concept with no bias in patient selection and beyond the standard so-called classical osteosarcoma patients; non-metastatic, extremity localised tumor.
The weakness is the limited data on treatment and the impact of less intensive chemotherapy on worse survival in non-children patients. Age is in general an adverse prognostic factor in oncology, as for osteosarcoma, the issue is whether poor outcome in older osteosarcoma patients is caused by improper chemotherapy or other treatment-related factors, that can be managed. Therefore, treatment related data including histological response to preoperative chemotherapy should be included to make the study relevant for the readers
Response: Thank you for your comment. We agree with your comment and noticed the lack of information about chemotherapy and histological response to chemotherapy. From the clinician’s perspective, an ideal cancer registry data should include pathologic clinical information (chemotherapy regimen and histologic response). However, a big data such as government managed national cancer registry data has pros and cons. The national cancer registry data provides data without any bias but does not provide detailed medical information. When we conceptualized this manuscript, we were aware of the limited nature of the national cancer registry data. Instead, we aimed to obtain osteosarcoma pictures of AYA.

Reviewer 2 Report
Thank you for submitting this paper which represents an epidemiological overview of a rare disease with a huge series. The vast majority of data are already known. Nevertheless there are few bias to be highlighted. Osteosarcoma involves a broad spectrum of subtypes in particular it is not specified whether only high grade OS were included nor a clear distinction between low grade and high grade tumours has been done in the analysis. This aspect could partially explain the results in OS (low grade are mostly reported in age 20-29) and the different approaches (surgery alone). Likewise there are no details on elderly OS which often are secondary OS with a poor prognosis.
I would also insert a reference to EURAMOS study which is the lattest randomized study on OS
Author Response
Osteosarcoma involves a broad spectrum of subtypes in particular it is not specified whether only high grade OS were included nor a clear distinction between low grade and high grade tumors has been done in the analysis. This aspect could partially explain the results in OS (low grade are mostly reported in age 20-29) and the different approaches (surgery alone).
Response: Thank you for the comment. We agree with you. Regrettably, information on tumor grades is not available from the Korean national cancer registry data. Therefore, we could not specifically analyze using grade information. We have added this variable in the limitation.
(Page 8, line 222)
moreover, there remains no information regarding tumor size, tumor grade, chemotherapy regimen, and histopathological response to preoperative chemotherapy.
Likewise, there are no details on elderly OS which often are secondary OS with a poor prognosis.
Response: Secondary osteosarcomas were defined as “cases that have been diagnosed and treated for other malignancies before the diagnosis of osteosarcoma.” There were 15 secondary osteosarcoma cases in the national cancer registry as shown in the table below. Cases according to ages were as follows: 1 in ≤9 year-olds, 2 in 10–14 year-olds, 4 in 15–39 year-olds, 5 in 40–64 year-olds, and 3 in those ≥65 years. Secondary osteosarcoma cases constituted a very limited proportion of the entire study cohort including the elderly, and therefore, it did not have a significant impact on the survival.
Survival rate (%)
Survival time (yr.) |
All patients |
Secondary OS |
Patients excluding Secondary OS |
|||||
Total |
0–9 |
10–14 |
15–39 |
40–64 |
≥65 |
|||
N=3017 |
N=15 |
N=3002 |
N=213 |
N=608 |
N=1303 |
N=603 |
N=275 |
|
2 |
76.26 |
51.33 |
76.38 |
83.93 |
85.53 |
83.68 |
63.23 |
43.29 |
5 |
61.92 |
42.78 |
62.01 |
77.54 |
73.47 |
68.14 |
47.12 |
24.98 |
7 |
59.26 |
34.22 |
59.38 |
75.12 |
70.26 |
65.67 |
44.63 |
22.19 |
10 |
56.87 |
34.22 |
56.98 |
73.83 |
67.32 |
63.66 |
41.31 |
18.94 |
I would also insert a reference to EURAMOS study which is the latest randomized study on OS.
Response: The EURAMOS trial has been added in the discussion and reference list.
(Page 7, lines 175-178)
The EURAMOS study also reported that adolescents (males: 13–17 years; females: 12–16 years) and adults (males: ≥18 years; females: ≥17 years) had inferior event-free survival than had children (males: 0–12 years; females: 0–11 years) [8].

Round 2
Reviewer 2 Report
Some concerns still remain but I think authors have tried to patch the limitations due to the registry